# Static Balance Digital Endpoints with Mon4t: Smartphone Sensors vs. Force Plate

**DOI:** 10.3390/s22114139

**Published:** 2022-05-30

**Authors:** Keren Tchelet Karlinsky, Yael Netz, Jeremy M. Jacobs, Moshe Ayalon, Ziv Yekutieli

**Affiliations:** 1Mon4t LTD., Tel-Aviv 6706057, Israel; keren@mon4t.com; 2The Academic College at Wingate Institute, Netanya 4290200, Israel; neyael@wincol.ac.il (Y.N.); m-ayalon@wincol.ac.il (M.A.); 3Hadassah Medical Center, Department of Geriatric Rehabilitation, Faculty of Medicine, Hebrew University of Jerusalem, Jerusalem 9103401, Israel; jacobsj@hadassah.org.il

**Keywords:** smartphone, mobile app, static balance, force plate, center of pressure, sway, digital endpoints

## Abstract

Static balance tests are conducted in various clinics for diagnosis and treatment adjustment. As a result of population aging, the accessibility of these tests should be increased, in the clinic, and for remote patient examination. A number of publications have already conducted static balance evaluations using the sensors embedded in a smartphone. This study focuses on the applicability of using smartphone-based balance assessment on a large scale while considering ease of use, safety, and reliability. The Mon4t^®^ app was used to acquire the postural motion using different smartphone devices, different smartphone locations, and various standing postures. The signals derived from the app were compared to the center of pressure displacement derived from a force plate. The results showed moderate to high agreement between the two methods, particularly at the tandem stance (0.69 ≤ *r* ≤ 0.91). Preliminary data collection was conducted on three healthy participants, followed by 50 additional healthy volunteers, aged 65+. The results demonstrated that the Mon4t app can serve as an accessible and inexpensive static balance assessment tool, both in clinical settings and for remote patient monitoring, which is key for enabling telehealth.

## 1. Introduction

Balance control and postural stability prevent the body from falling while performing a range of activities, and are therefore crucial for everyday tasks [1,2]. Furthermore, they are important indicators in physical and neurological assessment for the diagnosis, monitoring, and grading of disease progression, as well as treatment [3,4]. For example, gait impairment and postural instability are among the main features of Parkinson’s disease [5,6] or vestibular dysfunction [1]. The need for an accessible and quantitative system for static balance assessment was enhanced during the COVID-19 pandemic, where patients had to be monitored and examined from home. The importance of this system will continue to grow due to population aging [7,8,9,10].

Previous studies with the Mon4t^®^ app (by Montfort Brain Monitor, Tel-Aviv, Israel) have provided quantitative estimations of gait by conducting a test commonly used in the clinic, Timed Up and Go (TUG). Mon4t uses the integral tri-axial accelerometer, gyroscope, and magnetometer that are embedded in any standard smartphone to calculate gait parameters, such as TUG completion time, segment times (e.g., standing up, rotation), step length, step-to-step variability, and more. All extracted digital endpoints were found to be meaningful and correlated to physicians’ recorded completion time [11]. These data also correlated with the subject’s Unified Parkinson Disease Rating Scale [12], including psychiatric patients on neuroleptics that developed parkinsonism [13]. The Mon4t app was further validated against other standard methods available in clinics and motion laboratories for quantitative TUG analysis, e.g., stop-watch [11], force treadmill, wearable sensors, and a 3D motion capture camera system [14].

After validating the Mon4t app usability for gait assessment, this study focused on static balance, providing indicators for impaired balance and fall risk [1,4,15,16]. Traditionally, static balance is measured with clinical rating scales [17,18,19,20] or by a force plate (FP) [15,21,22], which can be found in motion, biomechanics, or rehabilitation labs [23]. First, the mechanical pressure applied by the subject’s feet is measured while standing on the FP. Then, a weighted average of all pressure points is calculated and defined as the center of pressure (COP), which correlates with the center of mass (COM) [24,25], indicating body stability [26].

In addition, there are several new technologies with high sensitivity in capturing balance, including a mobile balance board [10], inertial wearable sensors [27,28], optoelectronic systems [29], and a mobile pressure mat [30]. More recent studies reported that embedded smartphone accelerometers may also assess postural stability and perform balance sway analysis [31,32]. They showed moderate to high correlations with FP assessment, which is a significant contribution to smartphone-based balance assessment validation. Their findings showed that smartphone accelerometers can distinguish between older adults at low and high risk of falling during semi-tandem and tandem stances [31]. In addition, de Groote et al. [32] performed test–retest reliability on a large group of healthy adults and found that the sensitivity of a smartphone is smaller than that of an FP; therefore, more repetitions or longer trials are required when using a smartphone. Paillard et. al. [7] and Moral-Munoz et al. [33] provide an overview of other existing applications for balance assessment, most of which are not validated against the FP. However, to the best of our knowledge, the gold standard for quantitative balance assessment is the FP, and therefore we chose it as our reference, similarly to these recently published works.

Various parameters are commonly derived from the COP measurement to evaluate postural instability, mostly COP displacement, also known as the “sway” biomarker/endpoint [7,21,27,34]. Sway is recorded in the mediolateral (ML) and anterior–posterior (AP) directions while standing in different postures [15,27], such as neutral stance (NS), one-leg stance (OLS), tandem stance (TS), and sometimes during other actions such as lifting an arm [1,4]. Unfortunately, conventional balance evaluation systems are still not sufficiently easy to use, affordable, and mobile; they are thus inaccessible for most relevant subjects and physicians are forced to use subjective assessments [28,31].

In this paper, we first demonstrate the correlation between the raw data obtained from all sensors available in a smartphone and the raw data obtained from an FP. We believe that this was missing in previous studies and that it is essential to strengthen the confidence in using a smartphone and to identify the best smartphone-derived data to be used. We then examine the various aspects related to making a smartphone-based solution available and accessible on a large scale by comparing different devices, device locations, and standing postures. Our findings allow us to determine the preferred settings for conducting a smartphone-based balance assessment, taking into account accuracy, ease of use, and subject safety.

## 2. Materials and Methods

This study took place in the biomechanical laboratory at The Academic College at Wingate (Netanya, Israel), and included two main stages. In the first stage, the Mon4t app was compared to the FP at the raw signal level. In the second stage, the digital endpoints of the two methods were compared on a larger scale. Regarding the FP method, COP sway values were measured with a Kistler force plate (BioWare Version 5.3.0.7, 9281EA, Kistler, Germany) and normalized according to body weight. The Mon4t app provided sensor recording, data analysis, processing, and storage. Statistical analyses and signal processing were conducted with MATLAB (version 2018a, MathWorks, Natick, MA, USA). The similarity between the methods was evaluated using the Pearson correlation coefficient with a 95% confidence interval. The study was approved by The Hadassah Medical Center Ethics Committee, Hebrew University of Jerusalem, Israel (LocalTrial Identifier: 0074-19-HMO).

### 2.1. Participants 

At the raw signal level comparison, three healthy volunteers were examined in detail: One woman aged 72 and two men aged 45 and 72. Following the raw signal comparison, we conducted more extensive validation of the calculated sway endpoints within the two methods. A total of 50 community-dwelling healthy older adults (36 women and 14 men, aged 78.22 ± 6.25) participated in this study. All participants were able to walk and stand on one leg independently without aid and were cognitively intact with a low risk of falling. All participants gave their informed consent to inclusion in the study.

### 2.2. Experimental Design

We previously demonstrated a high correlation between different devices with different operating systems during gait performance. For the purpose of this study, at the first stage of raw signals, two smartphones were compared for 6 sessions × 3 axes (18 observations in total for the two devices), while standing in various static positions for 10 s. Two devices with different operating systems were used, iPhone X (Apple, Inc., Cupertino, CA, USA) and Galaxy S6 (Samsung, Seoul, South Korea). The first smartphone uses InvenSense MPU-6500™ and the second uses Bosch’s 6-Axis IMU. Both combine a 3-axis gyroscope and a 3-axis accelerometer.

The two devices were positioned on the sternum, one on top of the other, to ensure that their sensors are located in the same position, within 7.5 mm of one another (due to smartphone thickness).

Next, to examine the influence of various device locations on the body, the participants performed random static balance conditions and postural movements while standing on the FP for 30 s. In each session, two devices were worn at different locations on the subject while standing on the FP, as demonstrated in Figure 1. The two devices were worn on the subject’s upper back, and near the L4 vertebra height (as demonstrated in Figure 2b), or on the sternum and abdomen (as demonstrated in Figure 2a). Eight sessions were recorded for the devices on the abdomen and sternum and three sessions for the upper and lower back. Each session included the accelerometer and gyroscope sensors; therefore, the total number of observations was 16 and 6 for smartphones located on the front and the back, respectively. Each observation compared two FP outputs of COP in the AP and ML directions against 3 smartphone axes (defined in Figure 1a).

The second stage of the study compared the output digital endpoints, extracted from the raw signals of the two methods over a large population. In this stage, participants were asked to complete five balance trials of 30 s duration: NS, TS left, TS right, OLS left, and OLS right. It is important to note that there are not yet standardized test protocols for balance assessments in posturography and that the TS and the OLS do not represent natural daily stances. However, the three different stances we chose in the study are the most frequently used in the clinic for balance assessments, according to several studies [15,21,23,25,29,35]. Moreover, since these stances are not repeated daily and there is a possible learning effect, all sessions were recorded simultaneously by the force plate and smartphone. In this way, the effect of subject learning will be the same in both methods. The purpose of this paper is not to rate the learning curve applicable to these tests, but to demonstrate that the smartphone and FP are measuring the same things.

A resting period of 15 s was given between tests. The samples from the left and right legs were analyzed together, leading to a comparison of three balance assessments. To reduce the effect of shoe size and the subject’s weight, the FP was calibrated in advance for each participant before testing, to normalize the measured pressure. Then, the calibrated FP’s COP (center of pressure) output was compared to the smartphone outputs. Two participants did not perform the OLS test (right and left) due to safety concerns and one TS left sample was discarded because it was only partially recorded by the FP; therefore, their samples were excluded from the analysis.

### 2.3. Raw Signal Processing and Statistical Analysis

FP measurements included the standard COP coordinates in the XY plane, equivalent to COP ML and COP AP, sampled at 100 Hz. The two smartphones recorded the posture movement via the built-in sensors sampled at 100 Hz, in three axes: ML, AP, and superior-inferior (SI). In total, nine signals from each smartphone were obtained: acceleration m^2^/s × 3, angular velocity rad/s × 3, and three orientation angles—yaw, pitch, and roll, as defined in Figure 1a. Before signal processing and analysis, all raw signals (of both methods) were normalized and smoothed with a moving average filter (MATLAB function) to suppress the system’s noise. The filter’s span was set between 20 Hz to 200 Hz for FP and smartphone signals for small to large movements, respectively. The difference in the filter’s span was due to the concern that a wide span on a ‘silent’ signal deletes most of its information and characteristics.

In comparing the raw signals of the smartphone and the FP, they must be synchronized. To ensure this, each sample from the sensors of both methods of data acquisition had an output of a CSV file containing the time stamp vector with an accuracy of 10 ms. Using this, we were able to align the different signals in time and compare them. Moreover, cross-correlation (MATLAB function) was calculated for all pairs of signals of the two devices. The similarity between each pair of raw signal outputs of the different methods was determined by the obtained cross-correlation coefficient.

### 2.4. Digital Endpoint Extraction and Statistical Analysis

The smartphone endpoints were extracted by calculating the average and standard deviation of the time series vectors derived from 3 sensors (accelerometers, gyroscopes, and orientation angles), according to Equations (1) and (2):(1)$$\mathrm{Avg}.\;\mathrm{Acc}\backslash \mathrm{Gyro}\backslash \mathrm{Angle}\;\mathrm{SI}\backslash \mathrm{AP}\backslash \mathrm{ML}=\frac{1}{N}\overset{N}{\underset{n=1}{\sum }}X_{i}$$
(2)$$\mathrm{STD}.\;\mathrm{Acc}\backslash \mathrm{Gyro}\backslash \mathrm{Angle}\;\mathrm{SI}\backslash \mathrm{AP}\backslash \mathrm{ML}=\sqrt{\frac{\sum_{n=1}^{N}{\left(X_{n}-\bar{X}\right)}^{2}}{N-1}}$$

*N* denotes the vector (series) length, n is the sample index, and $\bar{X}$ represents the average of a series. Each sensor includes 3 output vectors for 3 different axes (SI, AP, and ML); i.e., the total output endpoints from the smartphone include 2 metrics × 3 sensors × 3 directions = 18 endpoints for 50 participants in each balance assessment (3 stances). The final 18 endpoints that were taken for the statistics are the average endpoints obtained from the two trials each participant performed.

FP’s postural sway endpoints are typically calculated from the COP movement in the AP and ML directions, based on the measured forces [27]. There are many metrics to calculate the sway parameter, which highlights the absence of standardization in clinical evaluation in general: for example, COP AP/ML velocity (mm/s), area (mm^2^), ML/AP mean displacement (mm), etc. [27,34]. In this study, three metrics were applied to the raw COP data. Two of them are commonly used: root-mean-square (RMS) using Equation (3), and the maximal displacement as defined in Equation (4), in ML and AP axis separately. The third metric is based on averaging the peak-to-peak COP displacements in time for AP and ML separately, as described in Figure 3, using Equation (5).
(3)$$\mathrm{COP}\;\mathrm{RMS}=\sqrt{\frac{1}{N}\overset{N}{\underset{n=1}{\sum }}COP{\left[n\right]}^{2}}$$
(4)$$\mathrm{COP}\;\mathrm{Max}=\underset{1\leq n\leq N}{\max}\left\{COP\left[n\right]\right\}$$
(5)$$\mathrm{COP}\;\mathrm{Mean}\;P2P\;=\frac{\sum_{i=1}^{k}\left|y_{i+1}-y_{i}\right|}{k-1}$$

COP refers to the coordination in the ML or AP direction (mm), and k is the number of “peaks” in the COP raw signal, marked in red in Figure 3b, and $y_{i}\;$ represents their coordination value in the ML or AP direction (mm).

In total, the FP method obtained 6 endpoints for each participant, 3 metrics × 2 directions (ML, AP). In addition, as was explained concerning the smartphone endpoints, these 6 endpoints are the average result of two trials performed by each participant in each balance assessment. 

The agreement between the obtained endpoints of the smartphone and the FP was determined using the Pearson correlation coefficient. The three balance stances were calculated between the 18 endpoints of the smartphone against the 6 endpoints of the FP. This allowed us to identify the most sensitive sensor relative to the COP signal.

## 3. Results

### 3.1. Raw Signal Results

In the first comparison of the raw signals between two smartphones with different operating systems, both placed at the sternum, the derived averaged Pearson coefficients were r=0.66±0.22, r=0.95±0.04, and r=0.83±0.30 for the accelerometers, gyroscopes, and orientation angles, respectively. An example of the raw signals derived from the gyroscopes of the two smartphones is displayed in Figure 4.

Next, the raw signals of the smartphone were compared to the FP 4 device locations. The results showed very small differences in the correlation coefficients (<0.01). The average Pearson correlation for all smartphone locations was around r≅0.64±0.21 (see Figure 5).

Figure 2 demonstrates the similarity between the raw signals of the FP and the smartphones at different locations during the NS position of one participant for 18 s. Figure 2a describes a session where the two smartphones were in the front of a subject while standing on the FP. ‘Device 1’, worn on the sternum, yielded correlation coefficients of 0.56 and 0.87 for the AP and ML directions, respectively. ‘Device 2’, worn on the abdomen, received 0.57 and 0.85 for the same trial. Figure 2b describes a session where the two smartphones were placed on the participant’s back while standing on the FP. The obtained correlation coefficients for ‘Device 1′, which was placed on the upper back, were 0.48 and 0.78 for the AP and ML directions, respectively, whereas ‘Device 2′, placed on the lower back, achieved 0.50 and 0.77 for the same trial.

Figure 6 qualitatively compares the raw signals of the FP and the smartphone at different locations and demonstrates the similarity between three static balance positions at different levels of difficulty.

The left column represents the obtained COP ML and AP signals under NS, TS, and OLS conditions (in Figure 6a,c,e, respectively) for 20 s on the FP. At the same time, the postural movement was recorded with a smartphone located on the sternum, as shown in the right column in Figure 6b,d,f for NS, TS, and OLS, respectively. Moreover, an example of failed OLS performance where a subject dropped his leg is presented in Figure 6g,h. All signals in Figure 2, Figure 4 and Figure 6 were aligned and processed according to the procedure in Section 2.3.

### 3.2. Digital Endpoint Results

Postural balance endpoints of 50 participants for three standing conditions were obtained with an FP and a smartphone (using the Mon4t app). The average results obtained with the FP, for AP COP RMS, were 6.9, 8.7, and 13.3 mm, and for ML COP RMS they were 6.7, 9.3, and 22.9 mm for NS, TS, and OLS, respectively. The smartphone’s average results for AP accelerations were 0.0065, 0.0089, and 0.013 m/s^2^, and for ML accelerations they were 0.0043, 0.011, and 0.0172 m/s^2^ for NS, TS, and OLS, respectively. Six comparison examples are presented in Figure 7 as scatter plots. Each plot describes the relationship between two selected endpoints which represent the posture static sway in the ML or AP direction, from the FP and the smartphone. The Pearson coefficient, shown for each plot, determines the degree of agreement between the two endpoints (see Section 2.4 for a detailed description of the analysis and calculations).

The resulting scatter plots (Figure 7) include several outliers, which can be divided into two groups. In the first group, the smartphone reports high inertial values and the FP reports low sway COP (relative to sampling average), whereas the second group reports the opposite. The first group describes cases in which the subject loses equilibrium, therefore resulting in large upper body movements to regain balance and very small COP displacements. In the second group, the subject loses balance and tries to regain balance by lowering, dropping, or moving aside at least one leg. This causes dramatic shifts in FP COP measurements but keeps the body straight and almost unchanged. Both cases indicate an imbalance, assuming the subject followed the instructions and made an effort to remain still. Table 1 summarizes all obtained Pearson correlation coefficients between the endpoint of the two methods, in accordance with Section 2.4.

## 4. Discussion

Objective measures of balance assessment in non-laboratory settings are necessary for clinical evaluation and remote patient monitoring [36,37,38]. Body-worn accelerometers were reported as potential devices for balance, gait, and fall risk assessments [2,27,29,34,39]. It follows that smartphones’ embedded inertial sensors may serve the same purpose, as demonstrated in [31,32]. The purpose of this work is to expand this potential application from research to practical use. The ease of use, safety, and reliability of the smartphone demonstrates it to be a proper static balance assessment tool. The Mon4t app was validated against the FP, the gold standard in static balance assessment to this day [7,21]. The study had two main stages: a detailed comparison between the raw signals of the two methods, followed by a large-scale comparison of the digital markers (endpoints) under various conditions.

The first stage confirmed that there is a strong correlation between the raw signals of different operating systems (iOS and Android), even for very subtle movements such as in static balance tests (see an example in Figure 4).

When examining the effect of smartphone position on its correlation with the FP in most studies related to measuring equilibrium, sensors are positioned in the lower back region next to the L4 vertebra [27,40,41], which is the closest point to COM. However, the sternum location captures the largest displacement for every degree of trunk motion because it has an anatomical amplification that allows for higher sensitivity to small COM changes [42,43]. Furthermore, in previous studies where the dynamic posture sway endpoints were derived using tri-axial accelerometers on the upper and lower trunk, it was confirmed that the results from the upper position have a higher specificity for detecting future falls [44]. The results of our study demonstrate that there is no significant difference between the four different locations compared—front, back, or lower or upper body. All four comparisons achieved a moderate Pearson coefficient, around 0.64, and none of them had any significant advantage over the others (Figure 5). The preferred location for smartphone placement is the sternum, as it allows for easy access to the clinician as well as the subject if the test is self-conducted. It is recommended to use a strap to hold the smartphone against the sternum. However, placement of the smartphone using one’s hands results in similar accuracy if severe hand tremors are not present.

The qualitative examinations of raw signals from both methods for different static balance conditions indicated that as the body position became less stable, more vibrations were measured in both methods (Figure 6). The easiest balance test, NS, yielded mostly constant signals from FP and smartphone sensors (Figure 6a,b), whereas the measured signals during the OLS balance test contained frequent changes, and much higher speeds, accelerations, and noise (Figure 6e,f). Although the two instruments measure different parameters (pressure vs. acceleration and angular velocity), signal distribution increases with balance test difficulty in all signals in a similar manner. Dropping the foot following balance loss is reflected by a sharp and sudden spike in the signal, adding noise and extremely high-frequency components relative to the baseline signal. In such cases, the COP signal of the FP is disrupted completely and cannot be compared to smartphone sensors. In general, the obtained qualitative results are consistent with previously reported results with FP or inertial sensors, whereas difficult balance assessments, such as OLS, presented higher COM displacements, velocities, frequencies, and accelerations [2,24,27,39].

In the second stage of the study, the endpoints of the two methods were compared. Two examples of the agreement between the obtained endpoints from the smartphone (gyroscope AP and accelerometer ML) and the two endpoints obtained from the FP for the three different balance assessments are presented in Figure 7. Table 1 lists all Pearson correlation coefficients between the endpoints of the two methods. Results were obtained for three calculation metrics for FP COP sway, for three standing postures, and the three types of smartphone sensors, as described in Section 2.4. When looking at the three postures, the gyroscope, accelerometer, and orientation angle presented correlation coefficients ranging from 0.4 < *r* < 0.88, 0.28 < *r* < 0.81, and 0.12 < *r* < 0.91, respectively. Analyzing the OLS and TS assessments without the NS results, all three sensors achieved similar moderate to high agreements: 0.61 < *r* < 0.88, 0.53 < *r* < 0.81, 0.59 < *r* < 0.91 for the gyroscope, accelerometer, and orientation angle, respectively.

Among the various standing postures, NS presented the lowest agreement of 0.12 ≤ *r* ≤ 0.60, OLS a moderate agreement of 0.53 ≤ *r* ≤ 0.71, and TS presented the highest agreement of 0.69 ≤ *r* ≤ 0.91. In both smartphone and FP, NS presented the smallest variance in all sway endpoint distributions among the three balance tests (Figure 7). This was also apparent in the raw signal results (Figure 6a,b), where the raw signals were almost constant, thus explaining the low correlation between the Mon4t app and the FP, as the signal lacks any distribution. In addition, TS distribution in the scatter plots is similar to OLS, (Figure 7b,e compared to Figure 7c,f). This was also noticed in the amplitudes and characterization of their raw signals (Figure 6c,d compared to Figure 6e,f), except for cases where the subject lost balance and dropped a leg during the OLS trial (Figure 6g,h). The duration of time in which a person can stand on one leg can serve as a standalone outcome for balance assessment [44], but in most cases, this will make the test duration too long for practical application. If the test outcome is the sway, dropping the leg in the middle of the test creates fewer comparative scores—for example, in the case of two subjects, one of whom maintains a stable OLS for 10 s compared to a subject who is unstable but balances for one second longer. Taking into account safety concerns, which are particularly important for remote patient monitoring, it can be concluded that TS posture should be preferred over OLS.

Another consideration is the possible effects of device inclination on the balance measurements while wearing the smartphone on the sternum. The output accelerations and velocities are influenced by the device orientation, according to their projection on the three defined axes, AP, ML, and SI. However, since the angles are relatively small, we did not see a significant effect on the correlation to the FP. Nevertheless, we also obtained the standard deviation of the yaw/pitch/roll signals, (as described in Section 2.4), which represent the changes in the orientation angle aligned to the absolute horizon, thus eliminating any device orientation effect. 

## 5. Conclusions

The Mon4t app demonstrated moderate correlations (average of r≅0.64) with FP at the raw signal level. Of the four device locations examined, placing the smartphone against the sternum yields a high sensitivity to body static balance motions, and allows for comfortable and friendly use of the smartphone, which is especially important for self-assessment. All three smartphone sensors—accelerometer, gyroscope, and orientation angle—demonstrated similar sensitivities to sway in the OLS and TS balance assessments (0.53 ≤ *r* ≤ 0.91).

Of the three standing postures, NS seems to offer the least amount of information. OLS may be too challenging for some subjects, and dropping a leg alters the results altogether, making the test incomparable to others. TS is more challenging than NS, thus spanning the range of sway, which enables the classification and evaluation of a person’s static balance. At the same time, TS is safer than OLS and has fewer invalid tests caused by leg drops, making TS the preferred stance for balance assessment out of the three.

While this study validated static balance assessment using smartphone technology, there are some limitations. First, this solution is suitable for people with technological literacy, who know how to operate a smartphone. Second, some smartphone devices (especially old models) may not have the needed sensors integrated into them; however, the Mon4t app checks for the presence of the required sensors before starting the test.

Telehealth and remote patient monitoring/treatment are among the most researched and evolving areas in medicine today. Given the increasing number of elderly people who use smartphones [45,46], applications such as Mon4t can enable an objective and self-conducted static balance assessment which is currently available only for in-clinic examination.

## Figures and Tables

**Figure 1 sensors-22-04139-f001:**
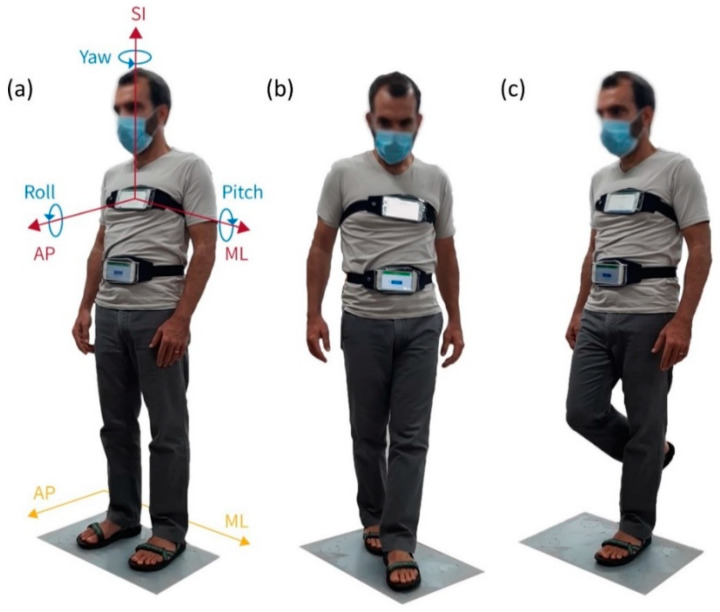
Subject standing on the force plate (FP) wearing two smartphones, one on the sternum and another on the abdomen, with straps, in three standing postures: (**a**) neutral stance (NS)—standing with spread legs; (**b**) tandem stance (TS)—standing with the heel of one foot against the toes of the other foot; (**c**) one-leg stance (OLS)—standing with one leg on the ground. The axis systems of the FP (yellow) and of the smartphones (red) are described in (**a**).

**Figure 2 sensors-22-04139-f002:**
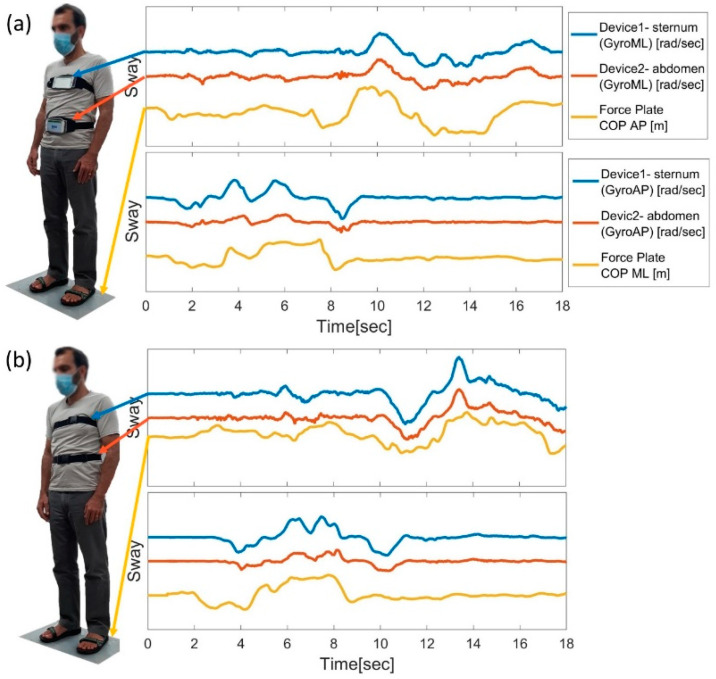
Mon4t vs. force plate (FP) raw signal comparison. Two methods captured the subject’s posture movement while standing in a neutral stance (NS) position. All graphs describe three normalized sensors for three different devices: Device 1 (blue), Device 2 (orange), and the FP (yellow). The two smartphones were placed on the subject’s (**a**) sternum and abdomen, or (**b**) on the upper and lower back, respectively.

**Figure 3 sensors-22-04139-f003:**
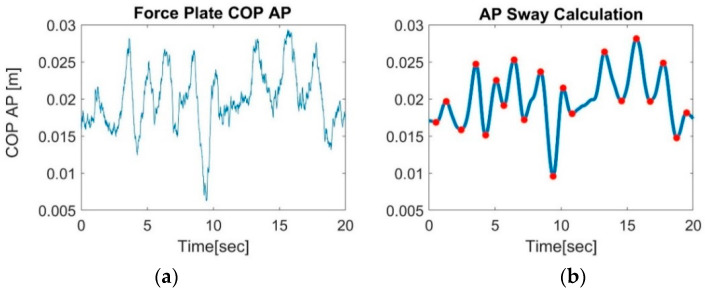
Sway calculations from the center of pressure (COP), measured using a force plate (FP). (**a**) Original FP’s COP output signal in the AP axis, recorded while standing on one leg. (**b**) Peak detection (red dots) over a smoothed COP signal from (**a**). For AP sway calculation, the average of all intervals between consecutive red peaks (peak-to-peak) was calculated.

**Figure 4 sensors-22-04139-f004:**
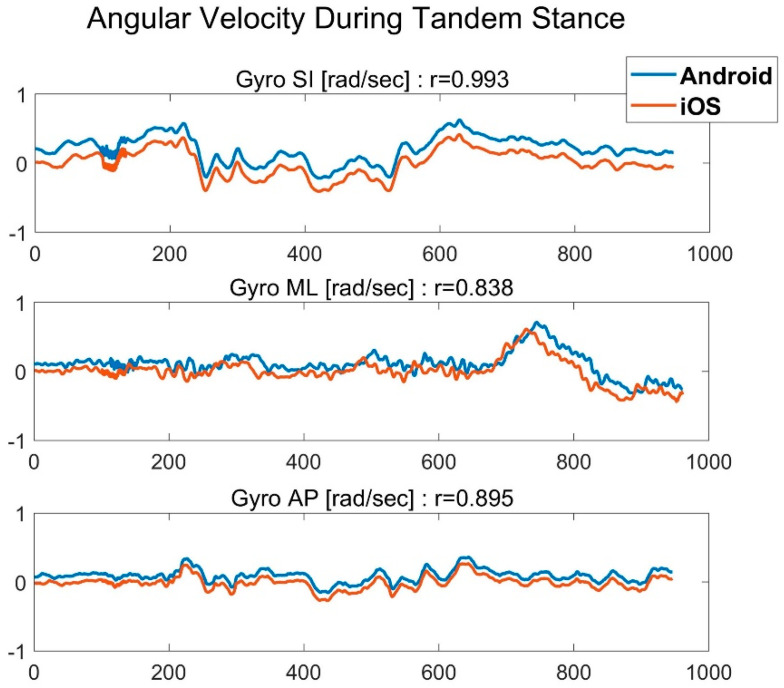
Cross-correlation between the raw signals, recorded with the gyroscope sensors of two smartphones with different operating systems, iOS (orange) and Android (blue), during tandem stance performance, with devices located on the participant’s sternum.

**Figure 5 sensors-22-04139-f005:**
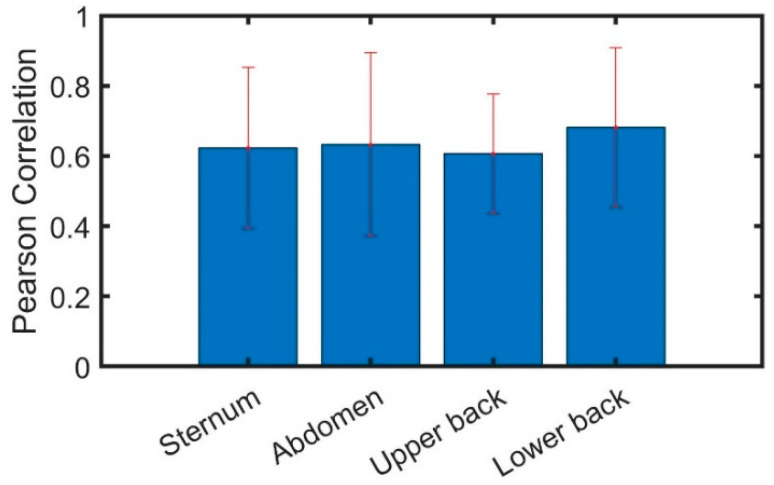
Average cross-correlation between the smartphone and the force plate across different device locations: (1) sternum, (2) abdomen, (3) upper back, and (4) lower back.

**Figure 6 sensors-22-04139-f006:**
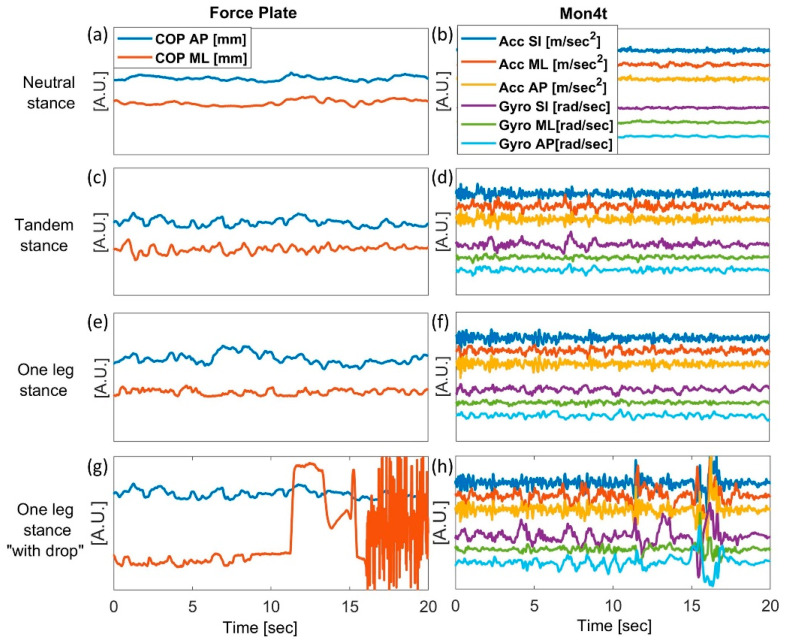
Force plate and Mon4t raw data across different static stances. The left column represents the normalized center of pressure coordination in the AP and ML directions. The right column represents the smartphone’s sensors: three axial accelerometers and gyroscopes, provided by the Mon4t app.

**Figure 7 sensors-22-04139-f007:**
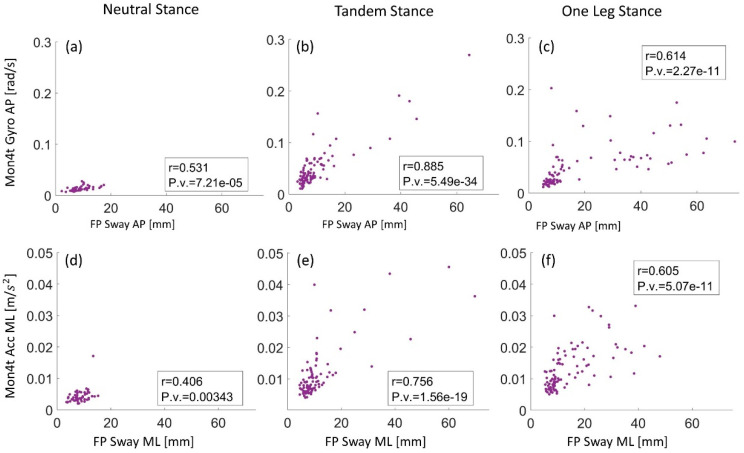
Mon4t vs. force plate (FP) within static sway endpoints. Comparison between the obtained static sway results of both methods in the AP (**a**–**c**) and ML (**d**–**f**) directions. (**a**,**d**) refer to sway performance during neutral stance, (**b**,**e**) during tandem stance, and (**c**,**f**) during one-leg stance.

**Table 1 sensors-22-04139-t001:** Pearson correlation coefficients between force plate (FP) and smartphone, within static sway endpoints. Three metrics were applied to the center of pressure (COP) to extract 6 sway endpoints from the FP (for the AP and ML directions). Smartphone sway endpoints were obtained from 3 types of sensors (accelerometer, gyroscope, and orientation angle). Pearson correlations presented with their obtained *p* values, * *p* ≤ 0.05, ** *p* < 0.01.; RMS = root mean square; AP = anteroposterior; ML = mediolateral.

		Neutral Stance		Tandem Stance		One Leg Stance	
FP Measures	Mon4t Measures	AP	ML	AP	ML	AP	ML
COP RMS	Mean Gyroscope	0.40 **	0.53 **	0.73 **	0.84 **	0.71 **	0.65 **
	SD Yaw/Pitch/Roll	0.12	0.41 **	0.82 **	0.91 **	0.68 **	0.70 **
	Mean Accelerometer	0.40 **	0.28 *	0.69 **	0.78 **	0.63 **	0.60 **
COP Mean P2P	Mean Gyroscope	0.60 **	0.46 **	0.80 **	0.88 **	0.61 **	0.70 **
	SD Yaw/Pitch/Roll	0.42 **	0.16	0.90 **	0.82 **	0.59 **	0.64 **
	Mean Accelerometer	0.31 *	0.41 **	0.81 **	0.76 **	0.53 **	0.60 **
COP Max	Mean Gyroscope	0.53 **	0.46 **	0.81 **	0.82 **	0.61 **	0.67 **
	SD Yaw/Pitch/Roll	0.52 **	0.17	0.83 **	0.87 **	0.65 **	0.72 **
	Mean Accelerometer	0.30 *	0.33 *	0.75 **	0.76 **	0.57 **	0.61 **

## Data Availability

The data supporting the reported results of this study are available from the corresponding author upon request.

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
