# Peer review of "Static Balance Digital Endpoints with Mon4t: Smartphone Sensors vs. Force Plate"

_sensors, 2022, doi:10.3390/s22114139_

Round 1
Reviewer 1 Report
The paper presents a evaluates a smartphone-based method for for static balance evaluation (mon4t) developped by the authors and compares its performance and results to those obtained by measuring center of pressure displacement derived from a force plate. In order to assess the reliability of the proposed method, different smartphone devices, different smartphone locations, and various standing postures have been analysed.
The experiments performed by the authors seem adequate, and they have been extended to a reasonable number of subjects (over 50 volunteers). Some points, however, are in my opinion poorly explained in the paper and should be clarified before the paper is published.
For example, the way data are acquired, transfered and stored for subsequent analysis isn't clear. How are the different data acquisition devices (smartphones, force plate) "synchronized"? (I mean, how do the authors guarantee that the data captured by the different devices share the same time reference, so that they can be correlated and plot in the same graphs?)
In my opinion, section 2, 3 and 4 should be reworked in order to better define the statistics presented in figures and tables in that section, and to improve the description on how they are computed. In particular, how is "sway" computed using acc and gyroscope shoud be better explained. Also how the values in table I are exactly computed? (i understand they are a kind of average of the pearson correlations values, but unsure if for all the tests or for all the subjects,....). I also am usure about the meaning of the Y-axis in subplot of fig7. Please define the statistics you present in a more clear and unequivocal way.
When comparing "raw results" between different mobile phone OS (Android vs iOS), I don't think the difference is caused by the OS, but by slight differences in location (since both phones can't be at the exact same location at the same time, and the inertial sensor is also probably placed in a different position inside each smartphone). Worst correlation between ACC measurements than between GYRO measurements could thus be caused by differences in the "default" inclination of each sensor. Thus, when comparing ACC to GYRO, I wonder if better results could be obtained by processing ACC data in a more sophisticated way (i.e. combining the thre axis to estimate inclination and then computing the "sway" by means of change of this inclination).
Are the data collected by the authors public/open? If not, I encourage the authors to release their data as a public opendata set in a repository such as ZENODO. This would make their research much more interesting for the scientific community and increase the overall interest of their work.
Reviewer 2 Report
It is a good train of thought for the authors to use the sensors built into the smartphone to evaluated the static balance. The subjects can easily complete the test actions according to the prompts, and can quickly obtain the evaluation results. This method has advantages in terms of cost, remote diagnosis and convenience. The key issue is the reliability of the posture swing data acquired by the smartphone. By comparing with the gold standard force plate, the authors proves that the combination of attitude sensors built in smartphones can meet the measurement requirements of static balance. And the sensitivity of the accelerometer and gyroscope data is compared, which is critical. Regarding the placement of the smartphone on the body, first, should the size and orientation of the smartphone be considered? Is it possible to consider more accurate positioning? Second, if the test is carried out by hand, how much will the shaking of the hand affect the test results?
The standard of standing posture used for balance testing has not been established in postography. The authors sets three stances in the article and recommends tandem stances. There is such a problem here. The tandem stance is not a stance close to the daily habit. Do you need to consider the learning effect of the subjects after multiple tests?
Specific issues:
- The moderate to high agreement mentioned in the summary section requires specific correlation coefficient values;
- The common methods for static balance assessment methods in paragraph 3 only mention clinical scales and force plate, however, there are many existing static balance assessment methods, which need to be more rigorously expressed.
Reviewer 3 Report
Report on paper "Static Balance Digital Endpoints with Mon4t: Smartphone Sensors vs. Force Plate" submitted by Karlinsky et al., for publication in Sensors (sensors-1667177).
The authors studied the applicability of using smartphone-based balance assessment on a large scale. The authors used Mon4t® app to acquire postural motion and demonstrated that it can serve as an accessible and inexpensive static balance assessment tool, both in clinical settings and for remote patient monitoring. The paper topic is though interesting, but it cannot be accepted in its present form and the authors must perform some modifications by addressing the following comments:
- In subsection 2.1, it will be interesting to discuss the specifications of the accelerometers and gyroscopes integrated in the two smartphones used for the tests.
- In subsection 2.3, what about the participant weight range, which has a significant impact on the static balance?
- It is not clear if the tests are repeated several times for each participant to ensure their validity.
- The author could explain why the smartphone gyroscopes demonstrated a higher sensitivity to sway than the accelerometers?
- What about the sensitivity of the proposed approach to temperature fluctuations?
- In the conclusion, the limitations of the proposed approach should be specified from a critical point of view, especially with regard to the used smartphone that could include sensors with lower performances.
- The quality of the figures must be enhanced.
Round 2
Reviewer 3 Report
The authors have addressed my comments sufficiently to recommend publication of the paper in its current form.